# The coupling coordination relationship between sports industry agglomeration and economic resilience in the Yangtze River Delta region

Hongyu Ma[ID]1*, Lingli Li2

1 School of Sports and Health, Nanning Normal University, Nanning, China, 2 School of Tourism and Culture, Nanning Normal University, Nanning, China

* mhy@nnnu.edu.cn

**Data Availability Statement:** All relevant data are within the manuscript and its Supporting information files.

## Abstract

With a rapidly growing sports industry worldwide, one may argue that sports industry agglomeration can play a crucial role in the economy of the sports industry. In particularly, the coupling linkage between sports industry agglomeration and economic resilience can be leveraged to promote both economic quality and efficiency. Based on data on three provinces and one city in the Yangtze River Delta region during the 2011–2020 period, this study uses the entropy-weighted TOPSIS method, coupling coordination degree model, and relative development models to explore the coupling coordination relationship between sports industry agglomeration and economic resilience in this region. The results show that: (1) Sports industry agglomeration shows an overall increasing albeit fluctuating trend with interprovincial differences. (2) Economic resilience has steadily increased, while the economic resilience kernel density curve generally shows a "dual peaks" trend. (3) The coupling coordination between sports industry agglomeration and economic resilience remains in a fluctuating, albeit coordinated state. These findings are relevant for the integration and high-quality development of the sports industry in the Yangtze River Delta region.

## 1. Introduction

In a post-pandemic world characterized by downward pressures on the global economy, trade frictions, and other impacts of the geo-political environment, most economies are experiencing heightened risks and uncertainties [1]. As such, the idea of resilience is again on the table to explain how actors should respond in the face of risk and uncertainty, and how they can minimize risk [2, 3]. In an uncertain domestic and external environment, if a country's economy has to remain healthy, it needs to possess economic resilience to continued changes in the external environment [4]. Scholars agree that economic resilience can, to a certain extent, enable China's economy to develop steadily by making timely adjustments in the face of external shocks, and provide a stable impetus for high-quality and high-efficiency development of its regional economies. Moreover, in the post-pandemic era, economic resilience can facilitate

**Funding:** The author(s) received no specific funding for this work.

**Competing interests:** The authors have declared that no competing interests exist.

a return to productive life and promote economic recovery [5, 6]. In line with this, the Central Committee of the Communist Party of China (CPC) recognized, at the national strategic level, that enhancing economic resilience, ensuring economic security, and building resilient cities are important issues in China's 14[th] and 23[rd] Five-Year Plans [7].

In particular, scholars have noted the necessity of empirically assessing the different associations between the sports industry and urban economic resilience. Some scholars have shown that the sports industry can contribute to urban economic resilience [8]. However, we need more in-depth analyses of this association in the context of macro-level value co-creation systems, such as "sports industrial agglomeration" [9], instead of the micro-level perspective adopted in prior studies [10]. Here, we focus on the Yangtze River Delta (YRD), which is an important region of China both economically and from the perspective of sports. In 2020, YRD's sports industry was as big as 1,052 billion yuan, accounting for 38.43% of the national sports industry, while its value added was 353 billion yuan. In the same year, the National Sports Bureau issued the "Several Opinions on the High-Quality Development of Sports Integration in the Yangtze River Delta Region". It outlines a focus on promoting regional planning convergence and coordinated development, advancing the implementation of the national strategy of national fitness, promoting the coordinated development of regional competitive sports, and creating models for the collaborative development of the sports industry and sports events. This reaffirms the importance of the sports industry and its economic status in the YRD region. Against this background, this study examines the coupling coordination relationship between sports industry agglomeration and economic resilience in the YRD region. Two research questions guide this study: What are the levels of sports industry agglomeration and economic resilience in the YRD region? Can the two form a coupling coordination relationship?

## 2. Theoretical foundations

### 2.1 The meaning of economic resilience and its components

The word resilience is derived from the Latin word *resilio*, meaning "to bounce back" [11]. Resilience is related to fundamental sociological concepts such as crisis, economic deprivation, sustainability, and vulnerability [12]. Holling introduced resilience into the ecological research framework in 1973 to analyze the ability of ecosystems to be maintained and repaired after natural or anthropogenic shocks [13]. Since then, it has evolved into a widely used term. In recent years, along with the impact of risks such as sudden public health events and the instability of the economic system, the concept has been gradually introduced into economics [1, 13]. Hill et al. argued that economic resilience means having the ability to prevent being thrown out of the previous equilibrium state due to an exogenous shock [14]. Martin observed that economic resilience is a kind of adaptive resilience, which mainly includes four dimensions: resistance, resilience, reconstruction, and renewal capacity [15]. Overall, scholars have reached a general consensus on the connotation of economic resilience and believe that the four-element theory can better explain.

### 2.2 Measuring sports industry agglomeration and economic resilience

Sports industry agglomeration is measured from the three dimensions of agglomeration scale, agglomeration speed, and agglomeration quality. Scholars also measure sports industry agglomeration at the industry level. Huang et al. measured the agglomeration of sporting goods manufacturing and sports service industries in 31 provinces, municipalities and autonomous regions in China [16]. Chen et al. measured the industrial agglomeration of the six major categories of sporting goods manufacturing industries in China from 2003 to 2009 [17].

Kim selected clusters of 13 sports industries to measure sports industry agglomeration based on North American Industry Classification System(NAICS) classification [8]. Typical agglomeration indicators used in these studies include industry concentration, location Gini coefficient, spatial separation index, location entropy, and Kernel density estimation [18, 19]. Meanwhile, Yao et al. used seven indicators belonging to the three aforementioned dimensions of sports industry agglomeration to explore the degree of coupled coordination (hereafter, coupling coordination degree) between sports industry agglomeration and regional economic growth [20].

Next, economic resilience is either measured using a core or multidimensional indicator. In the core indicator method, a particular macroeconomic indicator is selected, such as the number of employees or GDP, and the differences between them under the actual and counterfactual conditions are computed [13, 21]. For instance, Martin et al. selected an employment change indicator to measure economic resilience in the UK after four major recessions in the past 40 years [22]. He et al. also used the employment indicator to measure the economic resilience index of 230 cities in the shock-resistant and recovery-adjustment periods by stages [23]. The multidimensional indicator method involves constructing a multidimensional indicator system. For instance, Martin evaluated economic resilience based on the four dimensions of resistance, adaptability, organization, and resilience [24]. Wang et al. constructed an economic resilience evaluation indicator system with 14 indicators from the dimensions of resistance and resilience, adaptation and adjustment, and innovation and transformation [7].

## 2.3 The relationship between sports industry agglomeration and economic resilience

The degree of coupling characterizes the extent to which systems or elements interact and are interrelated. The degree of coordination characterizes the degree of benign interaction and coordinated development among systems or elements [25]. Scholars have undertaken preliminary discussions on the relationship between sports industry agglomeration and economic resilience, focusing on the following three aspects.

First, studies have focused on the impact of sports industry agglomeration on economic resilience. The sports industry is an activity-complex economy and one of the most comprehensive forms of an agglomeration economy. This is due to the joint positioning of a specific group of sports firms in the production chain to form an activity complex through forward (i.e., when a firm is located in the position of its customer firms) and/or backward linkages (i.e., when a firm is located in the position of its supplier firms) [26]. Consequently, the local sports industry, together with urban infrastructure and sports facilities, generates an activity-complex economy [27] with certain socioeconomic benefits and spillover effects, such as increased employment opportunities, improved quality of employment, improved infrastructure, social integration, sports participation, and public health benefits [8, 28]. Each city has its own unique social and material resources, which determines the different levels of development of its sports industry and urban economic resilience. However, the collaborative development of the sports industry and economic resilience can help avoid the pressures of "path dependence" on the regional economic system, and improve the sustainability and adaptability of the regional economy.

In addition, as a "sunrise industry" with strong pulling force, the sports industry can help in overcoming the geographical constraints between regions through industrial agglomeration. Moreover, it can accelerate the agglomeration and flow of capital, information, technology, talents, and other resources in the sports industry among regions. Meanwhile, the positive externality generated by the spatial agglomeration of resources promotes scale and

technology spillover effects. These influence the resistance, recovery, and transformation capacity of economic subjects [20]. However, we do not clearly understand the mechanisms through which sports industry agglomeration interacts with regional economic resilience is unclear.

Second, scholars have explored the impact of economic resilience on sports industry agglomeration. The sports industry is a large and complex system with external "driving force" and internal "vitality." Once a kinematic system is subjected to an external shock due to a reduction in external drive capacity or internal adaptability, it stagnates or collapses [29]. Therefore, economic resilience is essential for sports industry agglomeration, as the level of economic resilience can directly affect the level of specialization and diversification of the regional sports industry agglomeration. Furthermore, when the sports industry encounters an internal structural change or is affected by an external risk, the regional economic system can broaden the financing channels for the industry and increase the investment space to balance the industry's supply system. This can promote the stable development of the sports economy, and thus, help realize its specialization and diversification [30].

Finally, scholars have focused on the coupling coordination relationship between sports industry agglomeration and economic resilience. For the evolution of sports industry agglomeration, the sports industry and economic resilience must leverage the dynamic coupling system to realize mutual growth. Factors influencing the coupled and coordinated development of sports industry agglomeration and economic resilience in China include regional advantages, government behavior, value chain transmission, and environmental resource allocation [31]. Yang et al. constructed an evaluation indicator system for the sports industry and regional sustainable development, and showed improved coupling coordination between them using the coupling coordination degree model [32]. Clearly, studies demonstrate the role of sports industry agglomeration in improving economic resilience and vice versa, as well as the interaction between the two systems.

In summary, an extensive literature has examined the concept of economic resilience, and the influence between sports industry and economic resilience. However, few studies systematically explore the level of coordination and dynamic evolution of the coupling between sports industry agglomeration and economic resilience. Specifically, the following research gaps can be highlighted: (1) Regarding the research content, studies are mainly from the perspective of the interaction between sports industry agglomeration and economic resilience, and explore their internal relationship. Scholars rarely consider the bidirectional feedback of the coupling coordination degree. (2) Regarding research methodology, the coupling coordination degree model is commonly used to reflect the coupling coordination degree between sports industry agglomeration and economic resilience. However, the influence of the evolution of economic resilience on this coupling coordination in different regions has been ignored from the spatio-temporal perspective.

To address these gaps, this study uses data on three provinces and one city in the YRD region from 2011 to 2020, and explores the coupling coordination relationship between sports industry agglomeration and economic resilience based on the entropy-weighted TOPSIS method and coupling coordination degree models. This study's contributions are two-fold. First, this study explores the coupling coordination relationship between sports industry agglomeration and economic resilience in terms of their mutual influencing roles, and analyses the reasons for the differences between them. Second, this study analyses the evolutionary characteristics of economic resilience in the YRD region, deepens the understanding of the dynamics of the coupling coordination relationship between sports industry agglomeration and economic resilience, and proposes strategies for their coordinated development.

## 3. Methodology

### 3.1 Research methods

First, this study measures the degree of agglomeration of the sports industry using the location quotient (LQ). Second, the level of economic resilience is evaluated by the entropy-weighted TOPSIS method. Third, the coupling coordination degree of sports industry agglomeration and economic resilience is evaluated by the coupling coordinate degree model. Finally, the relative development coefficient of sports industry agglomeration and economic resilience is evaluated by the relative development degree model. Through the flowchart, we tried to find answers to two questions which are summarized in this paper [33]. Fig 1 shows the study's flowchart.

**3.1.1 Location quotient.** Industrial agglomeration can be measured using various measurement methods, such as spatial Gini coefficient, Herfindahl-Hirschmann index (HHI), Ellison-Glaeser index(E-G), and location entropy. Among these, the data availability needs are too demanding for the E-G index. Considering the data availability in this study's context and scientificity, the $LQ$ model is used to measure the degree of sports industry agglomeration in the YRD region [16]. $LQ$, also known as the "specialization rate," provides information on the degree of agglomeration of specific industries in a region [34]. It is also a technical tool to measure the spatial distribution of an industry in a country or region, thus reflecting the degree of specialization in a given industry sector [35]. Here, the $LQ$ of the sports industry is the ratio of the industry's added value in a province (or city) and its GDP (city) [19]:

$$LQ_{ij} = \frac{P_{ij}/P_{it}}{P_j/P_t} \tag{1}$$

$LQ_{ij}$ denotes the $LQ$ of the sports industry, $P_{ij}$ denotes the value added of the sports industry, and $P_{ij}$ denotes the value of GDP. In province (or city) $i$, $P_j$ denotes the value added of the sports industry and $P_t$ denotes the value of the province's (city's) GDP. When $LQ_{ij} > 1$, the

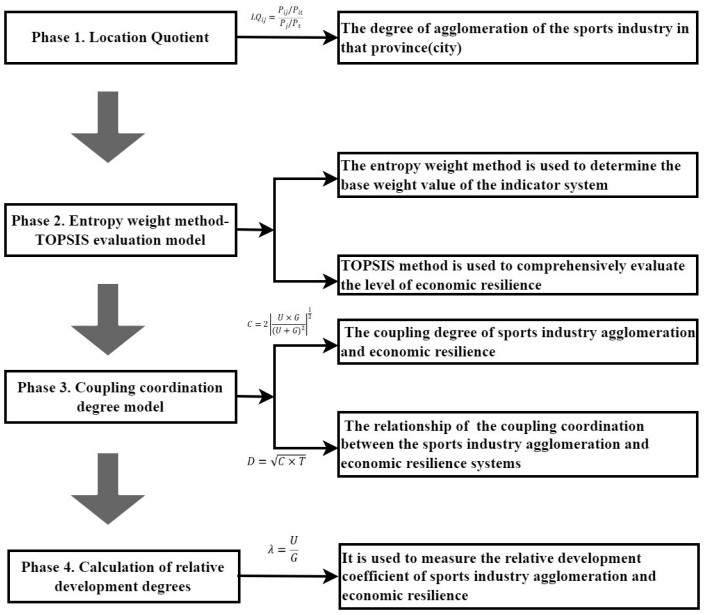

**Fig 1. Flowchart of the study.**

sports industry in province (or city) $i$, has a higher degree of agglomeration. When $LQ_{ij} < 1$, the sports industry in province (or city) $i$, has a relatively low agglomeration. When $LQ_{ij} > 2$, the sports industry has a very high level of agglomeration [36]. Thus, the entropy of the location of the sports industry in a province (or city) reflects the degree of agglomeration of the sports industry in that province (or city).

**3.1.2 Entropy-weighted TOPSIS method.** The entropy-weighted TOPSIS method combines the entropy weighting method and TOPSIS model. It can help reflect the evaluated value (hereafter, evaluation value) of economic resilience more objectively. The larger the evaluation value, the higher the region's economic resilience level; and vice versa [37]. Here, the entropy weight method is used to determine the base weight values of the indicator system, while the TOPSIS method is used to comprehensively evaluate the level of economic resilience.

(i) Fundamentals of the entropy weight method

The entropy weighting method is an objective weighting method to finalize the weights of indicators based on how different their values are during the indicator evaluation process. The validity of the amount of information reflected in the indicator's value is measured by the magnitude of the entropy value. The lower the entropy value, the higher the weight of the indicator [38]. The entropy method does not require expert intervention and can effectively eliminate human subjective influence [39]. Assuming m evaluated objects and n specific evaluation indicators, an initial evaluation matrix with m rows and columns can be formed as follows: $X = (X_{ij})$m*n, where $X_{ij}$ denotes the $j_{th}$ indicator value of the evaluation object. The specific steps for calculating the weights of the indicators are as follows:

Step 1. Standardize indicators. Since indicators have positive and negative attributes, different normalization algorithms are used to normalize the matrix $X = (X_{ij})$m*n and obtain the normalized matrix $P = (P_{ij})$m*n as follows:

$$P_{ij} = \frac{p_{ij} - minp_j}{maxp_j - minp_j} \tag{2}$$

$$P_{ij} = \frac{maxp_j - p_{ij}}{maxp_j - minp_j} \tag{3}$$

Step 2. Calculate the weight of each indicator:

$$P_{ij} = \frac{p_{ij}}{\sum_{i=1}^{m} p_{ij}} \tag{4}$$

Step 3. Calculate the entropy value of each indicator:

$$e_j = -K \sum_{i=1}^{m} lnp_{ij}, \ K = \frac{1}{ln_m} > 0 \tag{5}$$

Step 4. Calculate the coefficient of variation for the indicator:

$$h_j = 1 - ej(j = 1, 2, \ldots, n) \tag{6}$$

Step 5. Calculation the indicator weights:

$$W_j = \frac{h_j}{\sum_{j=1}^{n} h_j}, \, j = 1, 2, \ldots, n \tag{7}$$

(ii) Fundamentals of the TOPSIS model

Hwang and Yoon first proposed the TOPSIS model in 1981, ranking the evaluation objects according to their proximity to the idealized goal [40]. This method ranks the advantages and disadvantages by measuring the relative distance of the evaluation object from the optimal and worst programs [41]. This study adopts the TOPSIS model to evaluate and rank the level of sports industry agglomeration and economic resilience in the YRD region. It centers on determining a set of positive and negative ideal solutions as evaluation criteria, then calculating the closeness between them and the ideal solutions, and ranking the research objects according to the relative spatial distance [38]. The higher the degree of closeness (evaluation value), the higher the level of economic resilience of the province (or city). The formula is as follows:

Step 1. Construct a standardized decision matrix $(P_{ij})$ m*n and the weight matrix $W_{ij}$ (see below):

$$W_1 = \begin{bmatrix} w1 & 0 & \ldots & 0 \\ 0 & w2 & \ldots & 0 \\ \ldots & \ldots & \ldots & \ldots \\ 0 & 0 & \ldots & wn \end{bmatrix} \tag{8}$$

Step 2. Construct a weighted standardized decision matrix:

$$R = \left( r_{ij} \right)_{m*n} = \left( P_{ij} w_1 \right)_{m*n} \tag{9}$$

Step 3. Determine the positive and negative ideal solutions $S+$ and $S-$, respectively.

$S+ (S-)$ indicates that a specific indicator performs relatively well (poorly) on sports industry agglomeration and economic resilience in the province (or city) of the YRD region. It is computed as follows:

$$S^+ = \{r^+ | j = 1, 2, \ldots, n\} = \left\{ maxr_{ij} | j = 1, 2, \ldots, n \right\} \tag{10}$$

$$S^- = \{r^- | j = 1, 2, \ldots, n\} = \left\{ maxr_{ij} | j = 1, 2, \ldots, n \right\} \tag{11}$$

Step 4. Calculate the distance from the evaluation value vector to the positive and negative ideal solutions for each indicator:

$$d^+_i = \left[\sum_{j=1}^n |rij - r^+_j|^2\right]^{\frac{1}{2}} \tag{12}$$

$$d^-_i = \left[\sum_{j=1}^n |rij - r^-_j|^2\right]^{\frac{1}{2}} \tag{13}$$

Step 5. Calculate the relative closeness of the indicator, or the evaluation value, as follows:

$$C_i = \frac{d^-_i}{d^+_i + d^-_i}\,(i = 1, 2, \ldots, m) \tag{14}$$

Finally, the sports industry agglomeration and economic resilience level of each province (city) are comprehensively ranked by evaluation value. The larger the evaluation value, the higher the sports industry agglomeration and economic resilience level of the province (city), and vice versa.

**3.1.3 Coupling coordination degree model.** The coupling coordination degree model can reflect the function and structure of the system, especially the degree of mutual influence and interaction between different elements and systems. It includes the following indicators: degree of coupling ($C$), and the coupling coordination degree ($D$) [42]. The degree of coupling ($C$) characterizes the degree of mutual influence and interaction between systems. The coupling coordination degree ($D$) characterizes the degree of coordination between systems or between elements within systems [43]. Following Ye [43], this study constructs the following coupling coordination model to measure the coupling coordination relationship between sports industry agglomeration and economic resilience:

$$C = 2\left|\frac{U \times G}{(U + G)^2}\right|^{\frac{1}{2}} \tag{15}$$

$C$ ranges between [0,1]. A larger value indicates that the degree of interaction and mutual influence between sports industry agglomeration and economic resilience is higher. $U$ and $G$ represent the comprehensive indices of sports industry agglomeration and economic resilience, respectively.

$$T = \alpha \times U + \beta \times G \tag{16}$$

$T$ denotes the comprehensive coordination index between the sports industry ag-glomeration and economic resilience subsystems in the YRD region. $\alpha$ and $\beta$ are the weights of the two systems, respectively, and satisfy $\alpha + \beta = 1$. According to Ye [43] and in line with practice, this study considers that the two systems are equally interacting and influencing each other. Thus, $\alpha$ and $\beta$ equal 0.5.

$$D = \sqrt{C \times T} \tag{17}$$

$D$ ranges between [0,1]. The higher the coupling coordination degree, the stronger the

**Table 1. Criteria for types of the coupling coordination degree.**

| Typology | Dysfunctional decline category | | | | Excessive harmonization category | | Harmonized development category | | | |
|---|---|---|---|---|---|---|---|---|---|---|
| Level | Extreme disorder decline | Severe dysregulation recession | Moderate decline | Mild dysfunctional decline | Borderline dysfunctional decline | Barely coordinated integration | Elementary coordinated development | Intermediate coordinated development | Good coordinated development | Quality coordinated development |
| Interval | (0, 0.1] | (0.1, 0.2] | (0.2, 0.3] | (0.3, 0.4] | (0.4, 0.5] | (0.5, 0.6] | (0.6, 0.7] | (0.7, 0.8] | (0.8, 0.9] | (0.9, 1] |

coupling coordination between the sports industry agglomeration and economic resilience subsystems, and vice versa.

To more comprehensively analyze the reasons, this study adopts the criteria of Huang et al. [40] on the division of the coupling coordination degree and divides the synergy coefficients into three types of 10 levels shown in Table 1.

**3.1.4 Calculation of the relative development degrees.** To further clarify the internal constraints between the sports industry agglomeration and economic resilience subsystems, this study introduces the relative development degree model to measure the relative development coefficient of sports industry agglomeration and economic resilience as follows:

$$\lambda = \frac{U}{G} \tag{18}$$

$\lambda$ denotes the relative development degree, and $U$ and $G$ denote the composite indexes of sports industry agglomeration and economic resilience, respectively. Drawing on Li et al. [44], this study combines the results of the coupling coordination degree $D$ and relative development degree $\lambda$, and divides the synchronous relationship between the two systems into three levels. The details are shown in Table 2.

**Table 2. Classification of coordinated development of sports industry agglomeration and economic resilience.**

| Coupling coordination degree | Relative degree of development | Types and characteristics |
|---|---|---|
| $0 \le D \le 0.5$ | $0 \le \lambda \le 0.8$ | The level of sports industry agglomeration lags the level of economic resilience, and the two are highly antagonistic. |
| | $0.8 \le \lambda \le 1.2$ | The level of sports industry agglomeration is synchronized with the level of economic resilience, and the two are low antagonistic. |
| | $1.2 < \lambda$ | The level of sports industry agglomeration leads the level of economic resilience, and the two are in lowly antagonism. |
| | $0 \le \lambda \le 0.8$ | The level of sports industry agglomeration lags the level of economic resilience, and the two have a low degree of friction. |
| $0.5 \le D \le 0.7$ | $0.8 \le \lambda \le 1.2$ | The level of sports industry agglomeration is synchronized with the level of economic resilience, and the two have a high degree of friction. |
| | $1.2 < \lambda$ | The level of sports industry agglomeration leads the level of economic resilience, and the two have a high degree of friction. |
| | $0 \le \lambda \le 0.8$ | The level of sports industry agglomeration lags the level of economic resilience, with low coordination between the two systems. |
| $0.7 \le D \le 1$ | $0.8 \le \lambda \le 1.2$ | The level of sports industry agglomeration is synchronized with the level of economic resilience, and the two are highly antagonistic. |
| | $1.2 < \lambda$ | The level of sports industry agglomeration leads the level of economic resilience, and the two are in low coordination. |

## 3.2. Construction of sports industry agglomeration indicator system

According to the characteristics of sports industry agglomeration, and referring to the high-frequency indicators in Yao et al. [20] and Zhao et al. [42], we construct the evaluation system of sports industry agglomeration indexes. The sports industry agglomeration indexes contain three subordinate indexes for agglomeration scale, agglomeration speed, and agglomeration quality.

## 3.3. Construction of economic resilience indicator system

Following Lu et al. [1] and Zhu et al. [6], the evaluation indicator system of eco-nomic resilience is comprehensively constructed from the three dimensions of resistance and resilience, adaptation and adjustment, and innovation and transformation (see Table 3). Resilience and recovery are characterized by four indicators: GDP per capita, urban registered unemployment rate, and total exports and imports as a share of GDP. Adaptation and adjustment are characterized by two indicators: retail sales of consumer goods per capita and investment in fixed

**Table 3. Evaluation indicator system of sports industry agglomeration and economic resilience in Yangtze River Delta region.**

| Objective level | Criteria level | Indicator | Description | Attribute | Weighting |
|---|---|---|---|---|---|
| Sports industry agglomeration | Agglomeration scale (0.290) | Total value of sports industry (100 million) | Scale of sports industry development | + | 0.152 |
| | | Location Quotient | Degree of sports industry agglomeration | + | 0.138 |
| | Agglomeration speed (0.714) | Growth rate of output value of sports industry (%) | Development speed of sports industry | + | 0.138 |
| | | Value added of sports industry (100 million) | Development potential of sports industry | + | 0.149 |
| | | Growth rate of added value of sports industry (%) | Growth rate of sports industry | + | 0.137 |
| | Agglomeration quality (0.286) | Per capita value of sports industry (1,000 thousand) | Development status of sports industry | + | 0.147 |
| | | Share of sports industry in GDP (%) | Contribution of the sports industry to regional GDP | + | 0.139 |
| Economic resilience | Resistance and resilience (0.371) | GDP per capita (yuan) | Level of economic development | + | 0.075 |
| | | Urban registered unemployment rate (%) | Degree to which unemployment shocks the economic system | - | 0.076 |
| | | Total import and export as a share of GDP | Trade dependence | - | 0.073 |
| | | Per capita disposable income of urban residents/per capita disposable income of rural residents (yuan) | Regional urban-rural balance | - | 0.071 |
| | | Average wage of on-the-job workers (yuan) | Potential of social funds | + | 0.076 |
| | Adaptation and adjustment (0.298) | Per capita retail sales of social consumer goods (yuan) | Level of consumption | + | 0.075 |
| | | Per capita investment in social fixed assets (yuan) | Level of investments | + | 0.072 |
| | | Financial self-sufficiency rate | Fiscal self-adjustment capacity | + | 0.076 |
| | | Deposits of financial institutions in human currency (yuan) | Financial support capacity | + | 0.075 |
| | Innovation and transformation (0.331) | R&D expenditure as a share of GDP (yuan) | Intensity of financial investment in science, technology, and innovation | + | 0.075 |
| | | Patent authorization (pieces) | Innovation output level | + | 0.081 |
| | | Number of college students per 10,000 people (persons) | Innovation potential of social system | + | 0.077 |
| | | Index of advanced industrial structure (share of primary industry*1+ share of secondary industry*2+ share of tertiary industry*3) | Transformation capacity of economic system | + | 0.098 |

assets per capita. Finally, regional innovation and transformation capacity are characterized by two indicators: the proportion of R&D expenditure to GDP and number of patents granted.

Note that the urban registered unemployment rate, total exports and imports as a share of GDP, and disposable income per capita for urban and rural residents are negative indicators. The higher the urban registered unemployment rate, the greater the exposure of the regional economy to risk, or the less resilient the economy is. The share of total imports and exports in GDP reflects the trade dependence of the regional economy. A higher share means that the regional economy is more affected by international trade, or more susceptible to external shocks. This is not conducive for improving economic resilience [1]. A higher ratio of disposable income per capita between urban and rural residents implies a widening of the regional urban-rural imbalance. This is not conducive to the response of the economic system to external shocks and its automatic return to a state of stability.

To avoid the problem of non-comparability of the weights of the guideline-level indicators due to the different number of underlying indicators, this study refers to previous research. Specifically, the index weights of the criterion layer of the two systems are obtained by summing the index weights of the lower layer [37]. The evaluation indicator system is shown in Table 3.

### 3.4. Data sources

This study takes three provinces and one city in the YRD region (provinces: Jiangsu, Zhejiang, and Anhui; and city: Shanghai) as the object of study. The period of study is 2010–2020. Specific data are mainly derived from China Statistical Yearbook (2011–2021), Report on the Development of the Sports Industry in the YRD region (2014–2021), as well as the statistical yearbooks, statistical bulletins, and official websites of the sports bureaus of these geographies in the YRD region.

## 4. Results

### 4.1. Sports industry agglomeration in the Yangtze River Delta region

Table 4 lists the results for sports industry agglomeration calculated through the entropy-weighted TOPSIS method. Sports industry agglomeration level in the YRD region shows a

Table 4. Sports industry agglomeration in the Yangtze River Delta region, 2011–2020.

| Level of sports industry agglomeration | Shanghai | Jiangsu | Zhejiang | Anhui |
|---|---|---|---|---|
| 2011 | 0.229 | 0.377 | 0.333 | 0.244 |
| 2012 | 0.226 | 0.459 | 0.312 | 0.310 |
| 2013 | 0.277 | 0.463 | 0.352 | 0.224 |
| 2014 | 0.530 | 0.489 | 0.346 | 0.241 |
| 2015 | 0.455 | 0.562 | 0.435 | 0.317 |
| 2016 | 0.477 | 0.558 | 0.409 | 0.411 |
| 2017 | 0.505 | 0.596 | 0.426 | 0.321 |
| 2018 | 0.539 | 0.615 | 0.520 | 0.386 |
| 2019 | 0.521 | 0.659 | 0.521 | 0.371 |
| 2020 | 0.463 | 0.648 | 0.521 | 0.342 |
| Average | 0.422 | 0.543 | 0.543 | 0.317 |
| Average annual growth rate | 0.59% | 1.14% | 0.70% | 0.26% |

**Data source:** Report on the Development of the Sports Industry in the YRD region, National Report on the Total Scale and Added Value of the Sports Industry, and related websites.

general upward, albeit fluctuating trend. This fluctuation may be due to policy effects and various events. For instance, in 2014, the Chinese government introduced a policy to treat the sports industry as a sunrise industry. Consequently, sports industry agglomeration in the YRD region developed rapidly. Subsequently, due to the impact of the transformation and upgrading of the development of the sports industry, a slowdown in the rate of sports industry agglomeration followed [20]. Next, the COVID-19 pandemic in 2019 slowed down growth in the total value of the sports industry in the YRD region. However, sports industry agglomeration exhibited a slow downward trend. By province and city, Jiangsu and Zhejiang have the highest level of sports industry agglomeration. Shanghai has the second highest, while Anhui province is last. This indicates that the degree of independence of sports industry agglomeration in Jiangsu and Zhejiang provinces is high. Jiangsu province has the highest sports industry agglomeration growth rate, averaging more than 1% annually. The remaining provinces and cities grow slower at less than 1% annually on average. Clearly, an inter-provincial gap exists in sports industry agglomeration in the YRD region.

## 4.2. Economic resilience in the Yangtze River Delta region

Table 5 reports the results for economic resilience, which shows a general uptrend. In recent years, the YRD region has actively responded to the call of the integrated development of the region's sports industry. On the one hand, it is constantly optimizing sports industry structure, and promoting national fitness and sports consumption to push the high-quality development of this industry. On the other hand, it is constantly strengthening innovation in the sports industry, expanding the supply of sports products and services, and cultivating excellent sports talents. The goal is to accelerate the high-quality development of the sports industry and implement the strategy of comprehensively strengthening the country through sports. This series of policy measures has not only promoted the steady development of the sports industry economy in the YRD region, but also enhanced its resistance and resilience, adaptation and adjustment, and innovation and transformation, thereby improving the economic resilience level of the YRD provinces and city.

Jiangsu and Zhejiang provinces have the highest economic resilience. Shanghai is the second highest, while Anhui province has the lowest. However, Shanghai's economic resilience has increased faster at 2.28% annually on average. Meanwhile, Jiangsu province's resilience has grown at more than 1% on average. Finally, the economic resilience of the remaining two provinces has increased more slowly at less than 1% on average. Clearly, an inter-provincial gap exists in economic resilience in the YRD region, with a certain gradient effect.

To further clarify the evolutionary characteristics of economic resilience in the YRD region, the four years of 2011, 2014, 2017, and 2020 were selected for density estimation (as shown in Fig 2). The general trend of the skewed distribution of economic resilience has changed. With the passage of time, the center of the distribution curve of nuclear density in the YRD and interval of change have gradually shifted to the right. Further, the height of its main peak exhibits a gradual rise, followed by a decline, and then a significant rebound. Moreover, the

**Table 5. Results of the evaluation of economic resilience of the Yangtze River Delta region, 2011–2020.**

|  | 2011 | 2012 | 2013 | 2014 | 2015 | 2016 | 2017 | 2018 | 2019 | 2020 | Mean | Average annual growth rate |
|---|---|---|---|---|---|---|---|---|---|---|---|---|
| Shanghai | 0.344 | 0.398 | 0.417 | 0.434 | 0.464 | 0.490 | 0.509 | 0.547 | 0.565 | 0.596 | 0.477 | 2.28% |
| Jiangsu | 0.405 | 0.434 | 0.473 | 0.467 | 0.520 | 0.539 | 0.667 | 0.581 | 0.601 | 0.630 | 0.532 | 1.02% |
| Zhejiang | 0.374 | 0.406 | 0.435 | 0.425 | 0.474 | 0.499 | 0.519 | 0.553 | 0.574 | 0.560 | 0.482 | 0.77% |
| Anhui | 0.247 | 0.259 | 0.287 | 0.324 | 0.365 | 0.372 | 0.396 | 0.429 | 0.447 | 0.475 | 0.360 | 0.63% |

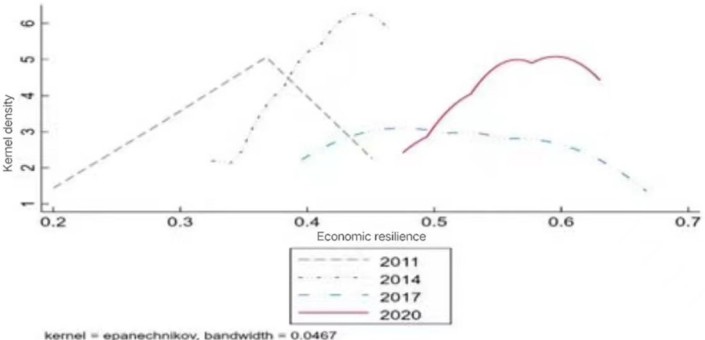

**Fig 2. Kernel density estimates of economic resilience in the Yangtze River Delta region.**

width of the curve increases and inter-regional convergence decreases. This indicates that economic resilience is constantly improving. However, the absolute gap in economic resilience and trend of divergence increase. This is closely related to the widening of the development gap between the three provinces and one city of the YRD region, and the movement of the distribution curve and YRD region's economic resilience.

In addition, from the viewpoint of the evolution of the wave crest, the height of the main peak in the YRD region has experienced the evolution process of "rising-declining-rising." Meanwhile, the width of the main peak has experienced the evolution process of "becoming smaller-bigger-smaller." This is generally characterized by the dynamic evolution process of the main peak's height fluctuating upwards and its width fluctuating downwards, and the position of the wave crests of the YRD region for each year is skewed toward the right side. This indicates that most economic resilience levels of the YRD region are in the middle-high state, while the absolute inter-regional differences tend to widen.

Specifically, in 2011, the kernel density curve showed a steep "A" peak, with a straight downward trend in the right-hand peak and economic resilience was at a relatively low level. Subsequently, by 2014, the kernel density curve gradually shifted to the right and height of the main peak rose sharply in the form of a curve. However, there were two weak side peaks on the left side of the kernel density curve, indicating that economic resilience in the YRD region showed weak multi-polarization, with increasingly obvious differentiation between provinces and cities. Finally, in 2017, the kernel density curve shifted significantly to the right and the height of the main peak fell sharply, presenting a gentle "flattening" bimodal distribution. The right tail of the curve lengthened year by year, the economic resilience of the high value area gradually became prominent, and the distribution of ductility exhibits a trend of broadening. This indicates that with the passage of time, the economic resilience of the relatively low level of provinces and cities gradually achieved the high-value area. However, the different provinces and municipalities in the region have different speeds of development and different scales such that the economic resilience gap between the provinces and municipalities in the region continues to widen. By 2020, the kernel density curve has increased significantly, peak value has continued to increase, and trend has gradually changed from "flattening" to "towering." Specifically, the distribution comprises one main and left peaks each. The height of the two peaks is comparable, but the distribution of ductility shows a convergence trend. This implies a weak polarization of economic resilience in the YRD region, but the gap has been narrowed to a certain extent.

Overall, during the study period of economic resilience in the YRD region, the peak of the wave was roughly "dual peaks" with the curve shifting to the right over the years. Further, the

distribution of extensibility is in a state of ups and downs. This shows that the level of economic resilience in the region is increasing. However, the regional gap in the level of economic resilience is generally in a widening trend and has not narrowed significantly. Clearly, if the economic resilience of the three provinces and one city in the YRD region is not further optimized is not optimized in the future, this regional disparity will inevitably be in a constant state of widening and narrowing, which is not conducive to the promotion of the implementation of the national strategy of promoting economic resilience in the YRD region.

Combining the results of Tables 4 and 5, the levels of sports industry agglomeration and economic resilience in the YRD region shows synchronization. This offers further evidence of the close endogenous relationship between them.

## 4.3. The coupling coordination of sports industry agglomeration and economic resilience in the Yangtze River Delta region

**4.3.1 Coupling coordination results.** Table 6 shows that the coupling coordination degree of the "sports industry agglomeration-economic resilience" composite system in the YRD region has generally increased. It has evolved from a near-disordered recession in 2011 to a coordinated development in 2020. However, weak inter-provincial differences exist. Specifically, the average coupling coordination degree of the composite system in Jiangsu province is the highest, reaching 0.733. This indicates an intermediate coordinated development stage. The average coupling degrees in Zhejiang province and Shanghai are next at 0.670 and 0.667, respectively. This indicates primary coordinated development. Finally, the average coupling degree in Anhui province is the lowest at 0.581. This indicates barely coordinated integration. Overall, sports industry agglomeration and economic resilience show obvious regional synchronous change trends, and the coupling coordination degree is better. This shows that economic resilience promotes sports industry agglomeration in YRD region to a certain extent, while sports industry agglomeration is also the engine for improving economic resilience.

**4.3.2 Coupling coordination types.** Next, we analyze the coupling coordination degree types. The development status of sports industry agglomeration in the YRD region relative to the level of economic resilience is analyzed using the relative development degree measure. Then, based on the differences in the levels of development of sports industry agglomeration and economic resilience, the coupling coordination degree can be determined. The results are shown in Table 7.

Among the 40 research samples from 2011 to 2020, 7 cases (17.5% of all cases) have lagging sports industry agglomeration. Synchronization is observed in the remaining 33 cases (82.5%). By year, in 2011, the proportion of lagging and synchronized sports industry agglomeration was 25% and 75%, respectively. This proportion remained unchanged with stable development in 2014. However, all of them developed into synchronized types in 2017. By 2020, the proportion of lagging and synchronized sports industry agglomerations was again 25% and 75%, respectively.

By the change trend of the type of coupling coordination degree, sports industry agglomeration and economic resilience are always in synergistic development in Jiangsu and Zhejiang provinces. The two systems go through the process of "friction-coordination." Meanwhile, sports industry agglomeration and economic resilience in Shanghai city and Anhui province are changing between lagging and synchronized sports industry agglomeration. The two systems in Shanghai go through the process of "coordination-friction-coordination," and those in Anhui Province go through the process of "antagonism-coordination-friction-coordination." Overall, Shanghai, and Jiangsu and Zhejiang provinces have good sports resource endowment, industrial foundation, market scale, and policy environment. Consequently, they can adapt

**Table 6. Coupling coordination measurement of sports industry agglomeration and economic resilience in the Yangtze River Delta region, 2011–2020.**

| | Shanghai | | Jiangsu | | Zhejiang | | Anhui | |
|---|---|---|---|---|---|---|---|---|
| | Coupling coordination degree | Coupling coordination stage | Coupling coordination degree | Coupling coordination stage | Coupling coordination degree | Coupling coordination stage | Coupling coordination degree | Coupling coordination stage |
| 2011 | 0.533 | Barely coordinated integration | 0.628 | Primary coordinated development | 0.597 | Barely coordinated integration | 0.497 | On the verge of dysfunctional decline |
| 2012 | 0.551 | Barely coordinated integration | 0.671 | Primary coordinated development | 0.599 | Barely coordinated integration | 0.535 | Barely coordinated integration |
| 2013 | 0.586 | Barely coordinated integration | 0.687 | Primary coordinated development | 0.628 | Primary coordinated development | 0.506 | Barely coordinated integration |
| 2014 | 0.696 | Primary coordinated development | 0.695 | Elementary coordinated development | 0.622 | Primary coordinated development | 0.531 | Barely coordinated integration |
| 2015 | 0.681 | Primary coordinated development | 0.738 | Intermediate coordinated development | 0.677 | Primary coordinated development | 0.587 | Barely coordinated integration |
| 2016 | 0.699 | Elementary coordinated development | 0.744 | Intermediate coordinated development | 0.675 | Primary coordinated development | 0.629 | Primary coordinated development |
| 2017 | 0.715 | Intermediate coordinated development | 0.796 | Intermediate coordinated development | 0.688 | Primary coordinated development | 0.600 | Primary coordinated development |
| 2018 | 0.740 | Intermediate coordinated development | 0.776 | Intermediate coordinated development | 0.735 | Intermediate coordinated development | 0.642 | Primary coordinated development |
| 2019 | 0.739 | Intermediate coordinated development | 0.792 | Intermediate coordinated development | 0.742 | Intermediate coordinated development | 0.641 | Primary coordinated development |
| 2020 | 0.727 | Intermediate coordinated development | 0.802 | Good coordinated development | 0.737 | Intermediate coordinated development | 0.638 | Primary coordinated development |
| Mean | 0.667 | Elementary coordinated development | 0.733 | Intermediate coordinated development | 0.670 | Elementary coordinated development | 0.581 | Barely coordinated integration |

and adjust themselves quickly when they suffer from external shocks, breaking the path dependence and opening up new growth paths in the long-term dynamic perturbations. Meanwhile, when they face internal pressures, they can actively adjust the structure of the sports industry and enhance their economic resilience. By contrast, the sports industry in Anhui Province has a relatively weak foundation and is a relatively single industry structure dominated by the sports manufacturing industry [45]. Hence, when they suffer from external shocks and internal pressures, sports industry agglomeration is more vulnerable and relies more on economic resilience to ensure its development.

## 5. Discussion

This study constructs a coupled coordination evaluation model of sports industry agglomeration and economic resilience, and investigates their coupled and coordinated development in the YRD region. Theoretically, there are goal convergence, reaction, and opposite reaction relationships between the sports industry and economic resilience, especially during special

 

**Table 7. Types of coupling coordination degree of sports industry agglomeration and economic resilience in the Yangtze River Delta region.**

| Year | Province, City | Coupling coordination degree | Relative degree of development | Types and characteristics |
|---|---|---|---|---|
| 2011 | Anhui | 0.497 | 0.988 | Economic resilience is synchronized with sports industry agglomeration, with low antagonism between the two systems |
| | Shanghai | 0.533 | 0.666 | Economic resilience leads sports industry agglomeration, the two have low coordination |
| | Jiangsu | 0.628 | 0.931 | Economic resilience is synchronized with sports industry agglomeration, and the two are highly coordinated |
| | Zhejiang | 0.597 | 0.890 | Economic resilience is synchronized with sports industry agglomeration, with a high degree of friction between the two systems |
| 2014 | Shanghai | 0.696 | 1.221 | Economic resilience is synchronized with sports industry agglomeration, with a high degree of friction between the two systems |
| | Jiangsu | 0.695 | 1.047 | Economic resilience is synchronized with sports industry agglomeration, with a high degree of friction between the two systems |
| | Zhejiang | 0.622 | 0.814 | Economic resilience is synchronized with sports industry agglomeration, with a high degree of friction between the two systems |
| | Anhui | 0.531 | 0.744 | Economic resilience leads sports industry agglomeration, low coordination between the two systems |
| 2017 | Shanghai | 0.715 | 0.992 | Economic resilience is synchronized with sports industry agglomeration, with high coordination between the two systems |
| | Jiangsu | 0.796 | 0.894 | Economic resilience is synchronized with sports industry agglomeration, with high coordination between the two systems |
| | Zhejiang | 0.688 | 0.821 | Economic resilience is synchronized with sports industry agglomeration, and the two are highly integrated. |
| | Anhui | 0.600 | 0.811 | Economic resilience is synchronized with sports industry agglomeration, and the two are highly integrated. |
| 2020 | Shanghai | 0.727 | 0.777 | Economic resilience is synchronized with sports industry agglomeration, and both highly coordinated |
| | Jiangsu | 0.802 | 1.029 | Economic resilience is synchronized with sports industry agglomeration, and both highly coordinated |
| | Zhejiang | 0.737 | 0.930 | Economic resilience is synchronized with sports industry agglomeration, and the two are highly coordinated |
| | Anhui | 0.638 | 0.720 | Economic resilience leads sports industry agglomeration, and both have low coordination |

**Note:** Due to table length constraints, four point-in-time cross-sectional data sets for 2011, 2014, 2017, and 2020 were selected for this study.

events such as epidemics and other extreme conditions [46]. However, the coupling between different provinces and one city in the YRD region may differ during the development of the sports industry due to both policy effects, and the development stage of the region's supply and social systems.

First, we first discuss the policy effects. The YRD region has a long history of industrial cooperation, starting with industrial radiation from Shanghai, which is gradually spreading to the two provinces of Zhejiang and Jiangsu, and finally to Anhui province participating in the division of labor and collaboration. However, this incremental approach to cooperation hinders the construction of a complete industrial system and fails to adequately ensure industrial integration in the YRD [47]. The empirical results show that the levels of sports industry agglomeration and economic resilience in Jiangsu, Zhejiang, and Shanghai are high compared to those in Anhui Province, while the degree of coordination between the sports industry agglomeration and economic resilience in Shanghai has not reached the level of quality coordination. To address this, the YRD region should attach importance to the top-level design of industrial policy and industrial planning, and strengthen the implementation, management,

and guarantee mechanisms for the high-quality integrated development of the YRD sports industry [28]. Meanwhile, the YRD "three provinces, one city, and one hospital" sports industry cooperative association, and YRD sports industry alliance need to actively participate in the integrated governance process.

Second, the degree of coupled and coordinated development of sports industry agglomeration and economic resilience is influenced by the stage of development of the region's supply and social systems. The results show that the sports industry agglomeration, economic resilience, and coupling coordination degree of Jiangsu and Zhejiang provinces show a consistent rising trend, while those of Shanghai and Anhui province show a general rising, albeit fluctuating trend. This is related to the stage of development of the regional supply and social systems among the three provinces and one city. Specifically, the technical and industrial space systems of the sports industry in Jiangsu and Zhejiang provinces are relatively perfect, and are driven by the industrial system of resource endowment, industrial scale, technical level, industrial chain, and other factors. This induces the formation of specialized and diversified spatial agglomeration forms of the sports industry, and realizes the optimal allocation of resources and technological spillover effect, thus helping enhance economic resilience. Meanwhile, Shanghai is focused on building a globally famous sports city; however, the scale of its sports industry is relatively low compared to Jiangsu and Zhejiang provinces. Further, the city's supply and social innovation systems are imbalanced; hence, the city has not realized its potential. Finally, the foundation of Anhui province's sports industry is relatively weak, its scale is small, and the industrial structure is relatively single and dominated by the sports manufacturing industry [47]. Consequently, its sports industry agglomeration and economic resilience are low. Hence, in the face of external shocks, sports industry agglomeration and economic resilience coupling coordination degree fluctuate. Therefore, when discussing the coupled and coordinated development of the YRD region as a whole, the coupled development stages of the supply and social systems among the provinces and cities should be considered. This can help to better promote the synergistic development of the sports industry and economic resilience. Some recommendations for doing this include strengthening the sports industry cooperation among the governments of the three provinces and one city, removing government barriers, promoting the optimal allocation of sports resources and free flow of production factors, and realizing the fluency and sharing of social capital, such as projects, information, technology, and resources [28].

A key issue in promoting the integration of the sports industry in the YRD region is how the coupling coordination degree can be used to promote the region's sports economy while combining local characteristics. Therefore, according to the development plan "Integration of Sports Industry in YRD Region," the path of integration and development of sports industry agglomeration and economic resilience should be outlined based on the consistency of goals, coupling and coordinated development, and mutual benefit of sharing resources.

## 6. Conclusions

First, sports industry agglomeration in the YRD region shows a general upward albeit fluctuating trend, although inter-provincial differences exist. Jiangsu and Zhejiang provinces and cities have the highest level of sports industry agglomeration. Shanghai is the second highest, while Anhui province is the lowest. While sports industry agglomeration in Jiangsu Province grows more than 1% annually on average, the remaining three provinces and cities grow below 1% annually on average.

Second, economic resilience in the YRD region has steadily increased, with the economic resilience kernel density curve generally showing a "dual peaks" trend. Similar to the pattern for sports industry agglomeration, Jiangsu and Zhejiang provinces have the highest levels of

economic resilience, followed by Shanghai and Anhui province. However, the economic resilience level of Shanghai has increased faster, with an average annual growth rate of 2.28%. Jiangsu province has an average annual growth rate exceeding 1%. Meanwhile, the economic resilience of the remaining two provinces has increased more slowly at below 1% annually. Thus, inter-provincial gaps exist in the level of economic resilience in the YRD region, with some gradient effects. Further, with time, the kernel density curve of economic resilience has shifted to the right. The height of the main peak gradually rises first, then falls, and finally rises again. Thus, it shows a "dual peaks" trend of change, with the width of the curve increasing and interregional convergence decreasing. This indicates that while economic resilience is continuously improving, the absolute inter-provincial gap has expanded substantially and divergent trends are being observed between provinces.

Third, the coupling coordination degree of sports industry agglomeration and economic resilience in the YRD region exhibits an upward, albeit fluctuating, overall coordinated state. The composite system of "sports industry agglomeration—economic resilience" appears to be a complex system with coupling and interaction characteristics, and shows the basic conditions for coordinated development. The coupling coordination degree of the composite system generally shows an upward trend and has transformed from being on the verge of dysfunctional recession to coordinated development in 2011. However, some inter-provincial differences exist. By the type of coupling coordination, Jiangsu and Zhejiang provinces have always maintained synergistic development and gone through "friction-coordination." Meanwhile, Shanghai and Anhui province are switching between sports industry agglomeration lagging and synchronizing with economic resilience. The sports industry agglomeration and economic resilience subsystems in Shanghai have both experienced "coordination-friction-coordination," while those in Anhui Province have experienced "antagonism-coordination-friction-coordination" to maintain a certain degree of stability. However, they have not yet reached high-quality coordination, and still need to be synergized and promoted by the involved stakeholders.

## Supporting information

**S1 File. Origin data.**
(XLSX)

**S2 File. The results of comprehensive evaluation model.**
(XLSX)

**S3 File. The results of coupling coordination degree model.**
(XLSX)

**S4 File. The results of relative development degrees.**
(XLSX)

## Author Contributions

**Conceptualization:** Hongyu Ma.

**Data curation:** Hongyu Ma.

**Formal analysis:** Hongyu Ma.

**Investigation:** Hongyu Ma.

**Supervision:** Hongyu Ma.

**Validation:** Hongyu Ma.

**Writing – original draft:** Hongyu Ma.

**Writing – review & editing:** Lingli Li.

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
