## [Decision Letter · Decision Letter 0]

31 Jan 2024

PONE-D-23-37782The coupling and coordination relationship between sports industry agglomeration and economic resilience in the Yangtze River Delta regionPLOS ONE

Dear Dr. Li,

Thank you for submitting your manuscript to PLOS ONE. After careful consideration, we feel that it has merit but does not fully meet PLOS ONE’s publication criteria as it currently stands. Therefore, we invite you to submit a revised version of the manuscript that addresses the points raised during the review process.

We look forward to receiving your revised manuscript.

Kind regards,

Lóránt Dénes Dávid, PhD

Academic Editor

PLOS ONE

Journal Requirements:

This study was supported by the National Social Science Foundation of China, Practical and Theoretical Research on Festivals and Sports in Guangxi to Firm Up the Community Con-sciousness of the Chinese Nation, Project (22BTY113)

5. We notice that your supplementary figures are uploaded with the file type 'Figure'. Please amend the file type to 'Supporting Information'. Please ensure that each Supporting Information file has a legend listed in the manuscript after the references list.

Reviewers' comments:

Reviewer's Responses to Questions

**Comments to the Author**

1. Is the manuscript technically sound, and do the data support the conclusions?

Reviewer #1: Yes

Reviewer #2: Yes

2. Has the statistical analysis been performed appropriately and rigorously? 

Reviewer #1: Yes

Reviewer #2: Yes

3. Have the authors made all data underlying the findings in their manuscript fully available?

Reviewer #1: Yes

Reviewer #2: Yes

4. Is the manuscript presented in an intelligible fashion and written in standard English?

Reviewer #1: No

Reviewer #2: Yes

5. Review Comments to the Author

Reviewer #1: General Comments:

The manuscript focuses on the spatial clustering of the sports industry and its economic relations, which aligns with my research interests. However, I have several concerns that I would like the authors to address in the revision:

1. The abstract lacks background information and fails to highlight the novelty of this study.

2. There have been numerous similar studies with the same research methods and topics. Therefore, it is unclear what justifies the publication of this manuscript. Have the authors reached any new conclusions or made any new discoveries?

3. There are markings in the main text that indicate either a revised version or an initial submission. Additionally, there are several instances of Chinese characters. Please clarify the status of the manuscript and ensure that it is free of non-English text.

4. The authors should verify the accuracy of the citations. The main text contains many error prompts such as "[Error! Reference source not found]."

5. The logic in the introduction is flawed, particularly when describing the research background and current situation. I am unsure how the mention of China-US trade and other elements effectively leads to the introduction of this study.

6. The authors must review their English writing.

7. Please condense the second section as much of the content appears to be common textbook explanations. The theoretical explanations are redundant, and there is a lack of review and critique of existing research.

8. Table 2 appears incomplete.

9. I suggest adding a flowchart and explanation in the methodology section.

10. It is recommended to update the references with more recent citations. I recommend the authors refer to: https://doi.org/10.7752/jpes.2013.01007.

Please address these concerns in the revised manuscript.

Reviewer #2: The study deals with a current topic, both the investigated area and the topic are relevant from the point of view of the research. The structure of the study is logical. It is recommended to present the correlations between the research questions and the applied methodology. The analysis of the performed tests is more descriptive, I recommend highlighting the correlations between the results. The obtained results are novel.

6. PLOS authors have the option to publish the peer review history of their article (what does this mean?). If published, this will include your full peer review and any attached files.

Reviewer #1: No

Reviewer #2: No

---

## [Author Response · Author response to Decision Letter 0]

13 Mar 2024

Dear Editors and Reviewers':

Thank you for your letter and for the reviewers' comments concerning our manuscript entitled “The coupling coordination relationship between sports industry agglomeration and economic resilience in the Yangtze River Delta region”. (Manuscript Number: PONE-D-23-37782). Those comments are all valuable and very helpful for revising and improving our paper, as well as the important guiding significance to our further researches. We have studied these comments carefully and have made correction which we hope meet with approval. Words in blue are the change I have made in the manuscript. The main corrections in the paper and the point-by-point responses to the Reviewers' comments are as following:

Response to Editors 

Response to requirement 1: Please ensure that your manuscript meets PLOS ONE's style requirements, including those for file naming.

Response: In accordance with PLOS ONE's style requirements, we have adjusted the formatting of the abstract, introduction, figures, and reference, and have upload this letter as a separate file labeled "Revised Manuscript with Track Changes" , and upload the other letter as a separate file labeled "Manuscript".

Response to requirement 2: Did you know that depositing data in a repository is associated with up to a 25% citation advantage. If you’ve not already done so, consider depositing your raw data in a repository to ensure your work is read, appreciated and cited by the largest possible audience. You’ll also earn an Accessible Data icon on your published paper if you deposit your data in any participating repository.

Response: I’d like to deposit the data in any participating repository, and we have uploaded the data as required.

Response to requirement 3: Please provide an amended statement that declares all the funding or sources of support (whether external or internal to your organization) received during this study. “There was no additional external funding received for this study.” in your updated Funding Statement. Please include your amended Funding Statement within your cover letter. 

Response: We have decided to delete the grant description in the manuscript for the National Social Science Foundation project "Practical and Theoretical Research on Guangxi Festivals and Sports on Enhancing the Chinese National Community Consciousness" (22BTY113). This is because during the writing of the article, the funders mentioned above were not involved in the research design, data collection and analysis, decision to publish, or writing of the manuscript. Misunderstandings due to students being first-time contributors and not understanding the funding statement. 

Response to requirement 4: PLOS requires an ORCID ID for the corresponding author in Editorial Manager on papers submitted after December 6th, 2016. Please ensure that you have an ORCID ID and that it is validated in Editorial Manager.

Response: We have updated and verified the ORCID ID account as required.

Response to requirement 5: We notice that your supplementary figures are uploaded with the file type 'Figure'. Please amend the file type to 'Supporting Information'. Please ensure that each Supporting Information file has a legend listed in the manuscript after the references list.

Response: We have made the required adjustments.

Response to requirement 6: Please review your reference list to ensure that it is complete and correct. If you have cited papers that have been retracted, please include the rationale for doing so in the manuscript text, or remove these references and replace them with relevant current references. Any changes to the reference list should be mentioned in the rebuttal letter that accompanies your revised manuscript. If you need to cite a retracted article, indicate the article’s retracted status in the References list and also include a citation and full reference for the retraction notice. 

Response: We are very sorry for our careless mistake and we have revised the reference, full details of the references were listed in the file of the“Revised Manuscript with Track Changes”, in line 602-769.

Response to Review #1

We are extremely grateful to Referee for pointing out the problem in this paper. As your suggestion, we have made relevant corrections in the paper and response to your comments.

1. The abstract lacks background information and fails to highlight the novelty of this study.

Response: Thanks for reviewer’s comments, we have revised the second section part of the manuscripts as “With a rapidly growing sports industry worldwide, one may argue that sports industry agglomeration can play a crucial role in the economy of the sports industry. In particularly, the coupling linkage between sports industry agglomeration and economic resilience can be leveraged to promote both economic quality and efficiency”, in line 10-13.

2. There have been numerous similar studies with the same research methods and topics. Therefore, it is unclear what justifies the publication of this manuscript. Have the authors reached any new conclusions or made any new discoveries?

Response: Thanks for reviewer’s comments, and we have made correction according to the Reviewer’s comments.

Firstly, in terms of the literature review, in summary, an extensive literature has examined the concept of economic resilience, and the influence between sports industry and economic resilience [27]. However, few studies systematically explore the level of coordination and dynamic evolution of the coupling between sports industry agglomeration and economic resilience. Specifically, the following research gaps can be highlighted: (1) Regarding the research content, studies are mainly from the perspective of the interaction between sports industry agglomeration and economic resilience, and explore their internal relationship. Scholars rarely consider the bidirectional feedback of the coupling coordination degree. (2) Regarding research methodology, the coupling coordination degree model is commonly used to reflect the coupling coordination degree between sports industry agglomeration and economic resilience. However, the influence of the evolution of economic resilience on this coupling coordination in different regions has been ignored from the spatiotemporal perspective. 

To address these gaps, this study uses data on three provinces and one city in the YRD region from 2011 to 2020, and explores the coupling coordination relationship between sports industry agglomeration and economic resilience based on the entropy-weighted TOPSIS method and coupling coordination degree models. This study's contributions are two-fold. First, this study explores the coupling coordination relationship between sports industry agglomeration and economic resilience in terms of their mutual influencing roles, and analyses the reasons for the differences between them. Second, this study analyses the evolutionary characteristics of economic resilience in the YRD region, deepens the understanding of the dynamics of the coupling coordination relationship between sports industry agglomeration and economic resilience, and proposes strategies for their coordinated development. In line 162-184.

Secondly, in terms of conclusion, By province and city, Jiangsu and Zhejiang have the highest level of sports industry agglomeration. Shanghai has the second highest, while Anhui province is last. This indicates that the degree of independence of sports industry agglomeration in Jiangsu and Zhejiang provinces is high. In line 353-356. Jiangsu and Zhejiang provinces have the highest economic resilience. Shanghai is the second highest. Combining the results of Tables 5 and 6, the levels of sports industry agglomeration and economic resilience in the YRD region shows synchronization. This offers further evidence of the close endogenous relationship between them, in line431-433. The coupling coordination degree of the "sports industry agglomeration-economic resilience" composite system in the YRD region has generally increased. It has evolved from a near-disordered recession in 2011 to a coordinated development in 2020. However, weak inter-provincial differences exist. Specifically, the average coupling coordination degree of the composite system in Jiangsu province is the highest, reaching 0.733. This indicates an intermediate coordinated development stage. The average coupling degrees in Zhejiang province and Shanghai are next at 0.670 and 0.667, respectively. This indicates primary coordinated development. Finally, the average coupling degree in Anhui province is the lowest at 0.581. This indicates barely coordinated integration. Overall, sports industry agglomeration and economic resilience show obvious regional synchronous change trends, and the coupling coordination degree is better. In line 441-451. 

According to the above explanations, it's necessary to do this research.

3. There are markings in the main text that indicate either a revised version or an initial submission. Additionally, there are several instances of Chinese characters. Please clarify the status of the manuscript and ensure that it is free of non-English text.

Response: We are very sorry for our careless mistake and we have revised the markings, Chinese characters, non-English text in the part of the manuscripts.

4. The authors should verify the accuracy of the citations. The main text contains many error prompts such as "[Error! Reference source not found]."

Response: We are very sorry for our careless mistake and we have revised the reference in the part of the manuscripts as follows: 

1. Lu H, Zhang C, Jiao L, Wei Y, Zhang Y. Analysis on the spatial-temporal evolution of urban agglomeration resilience: A case study in Chengdu-Chongqing Urban Agglomeration, China. International Journal of Disaster Risk Reduction, 2022;79:103167. https://doi.org/10.1016/j.ijdrr.2022.103167

2. Beichler S-A, Hasibovic S, Davidse B-J, Deppisch S. The role played by social-ecological resilience as a method of integration in interdisciplinary research. Ecology and Society. 2014;19:(3). https://dx.doi.org/10.5751/ES-06583-190304

3. Zhang J. Spatial effects of tourism development on economic resilience: an empirical study of Wenchuan earthquake based on dynamic spatial Durbin model. Natural Hazards.2023;115(1): 309-329. https://doi.org/10.1007/s11069-022-05556-9

4. Li L, Huang X. The Latest Research Progress of Regional Economic Resilience 

in China. In 2022 7th International Conference on Social Sciences and Economic Development, 2022,1745-1750. https://creativecommons.org/licenses/by-nc/4.0/

Full details of the references were listed in the manuscripts, line 602-769.

5. The logic in the introduction is flawed, particularly when describing the research background and current situation. I am unsure how the mention of China-US trade and other elements effectively leads to the introduction of this study.

Response: Thanks for reviewer’s comments, we mentioned the research background and current situation mainly because “during the study period, China's domestic and international situation underwent profound and complex changes due to the post-pandemic world characterized by downward pressures on the global economy, China–US trade friction, and other impacts of the external environment, which have significantly increased the uncertainty and risk of China's economic and social development, and such impacts had the greatest magnitude of the impacts on the YRD region. Therefore, this study considers the general background of the normalization of epidemic prevention and the double impact of the external environment such as the post-pandemic world characterized by downward pressures on the global economy and the China–US trade friction, and it is necessary to enhance the level of economic resilience of the Yangtze River Delta region to cope with the impact of the external environment.”

6. The authors must review their English writing.

Response: Special thanks to you for this comment. As your suggestion, we have proofread the English writing accordingly. Words in blue are the change I have made in the manuscript.

7. Please condense the second section as much of the content appears to be common textbook explanations. The theoretical explanations are redundant, and there is a lack of review and critique of existing research.

Response: Thanks for reviewer’s comments, we have revised the second section part of the manuscripts as follows: 

Firstly, the author reduce the elaboration on the theoretical explanations of economic resilience.

Despite the growing importance of the idea of economic resilience, the concept has not been carefully defined or measured. Moreover, understanding of the concept of economic resilience is an important prerequisite for studying it. 

Secondly, the authors add the review and critique of existing research. we have added review and critique of existing research part of the manuscripts as “In summary, an extensive literature has examined the concept of economic resilience, and the influence between sports industry and economic resilience [27]. However, few studies systematically explore the level of coordination and dynamic evolution of the coupling between sports industry agglomeration and economic resilience. Specifically, the following research gaps can be highlighted: (1) Regarding the research content, studies are mainly from the perspective of the interaction between sports industry agglomeration and economic resilience, and explore their internal relationship. Scholars rarely consider the bidirectional feedback of the coupling coordination degree. (2) Regarding research methodology, the coupling coordination degree model is commonly used to reflect the coupling coordination degree between sports industry agglomeration and economic resilience. However, the influence of the evolution of economic resilience on this coupling coordination in different regions has been ignored from the spatiotemporal perspective. 

To address these gaps, this study uses data on three provinces and one city in the YRD region from 2011 to 2020, and explores the coupling coordination relationship between sports industry agglomeration and economic resilience based on the entropy-weighted TOPSIS method and coupling coordination degree models. This study's contributions are two-fold. First, this study explores the coupling coordination relationship between sports industry agglomeration and economic resilience in terms of their mutual influencing roles, and analyses the reasons for the differences between them. Second, this study analyses the evolutionary characteristics of economic resilience in the YRD region, deepens the understanding of the dynamics of the coupling coordination relationship between sports industry agglomeration and economic resilience, and proposes strategies for their coordinated development”, in line 162-184.

8. Table 2 appears incomplete.

Response: Thank you for your careful check. We are very sorry for our negligence.

Table 2 is partially hidden and has been reformatted. In addition, the degree of coupled coordination between sports industry agglomeration and economic resilience is also based on previous studies, and it can be divided into 10 subtypes. The criteria for types of the degree of coupling coordination is shown in Table 2, in line 289.

9. I suggest adding a flowchart and explanation in the methodology section.

Response: Thank you for reviewers’ opinions, we have added a flowchart and explanation in the methodology section of the manuscript as “First, this study measures the degree of agglomeration of the sports industry using the location quotient (LQ). Second, the level of economic resilience is evaluated by the entropy-weighted TOPSIS method. Third, the coupling coordination degree of sports industry agglomeration and economic resilience is evaluated by the coupling coordinate degree model. Finally, the relative development coefficient of sports industry agglomeration and economic resilience is evaluated by the relative development degree model. Through the flowchart, we tried to find answers to two questions which are summarized in this paper[37]. Figure 2 shows the study's flowchart. ”, in line 187-194.

In addition, we have also updated the

---

## [Decision Letter · Decision Letter 1]

26 Mar 2024

PONE-D-23-37782R1The coupling  coordination relationship  between the two systems in the Yangtze River Delta regionPLOS ONE

Dear Dr. Ma,

Thank you for submitting your manuscript to PLOS ONE. After careful consideration, we feel that it has merit but does not fully meet PLOS ONE’s publication criteria as it currently stands. Therefore, we invite you to submit a revised version of the manuscript that addresses the points raised during the review process.

We look forward to receiving your revised manuscript.

Kind regards,

Lóránt Dénes Dávid, PhD

Academic Editor

PLOS ONE

Journal Requirements:

Reviewers' comments:

Reviewer's Responses to Questions

**Comments to the Author**

1. If the authors have adequately addressed your comments raised in a previous round of review and you feel that this manuscript is now acceptable for publication, you may indicate that here to bypass the “Comments to the Author” section, enter your conflict of interest statement in the “Confidential to Editor” section, and submit your "Accept" recommendation.

Reviewer #1: All comments have been addressed

Reviewer #3: All comments have been addressed

2. Is the manuscript technically sound, and do the data support the conclusions?

Reviewer #1: Yes

Reviewer #3: Yes

3. Has the statistical analysis been performed appropriately and rigorously? 

Reviewer #1: Yes

Reviewer #3: Yes

4. Have the authors made all data underlying the findings in their manuscript fully available?

Reviewer #1: Yes

Reviewer #3: Yes

5. Is the manuscript presented in an intelligible fashion and written in standard English?

Reviewer #1: Yes

Reviewer #3: Yes

6. Review Comments to the Author

Reviewer #1: Thank you for providing the revised manuscript. The author has addressed all of my concerns, and I now recommend the publication of this manuscript.

However, I noticed a minor writing issue. For instance, there are several instances of "between the two" in this manuscript, which seems informal and incomplete. I suggest modifying it to "between the two systems" or using other pronouns or abbreviations.

Furthermore, as someone equally interested in system coupling and the Yangtze River Basin, I recommend that the author consider referencing and citing https://doi.org/10.3389/fevo.2023.1148868

Reviewer #3: The authors have addressed all the comments provided in the first round of review. Therefore, the manuscript can be accepted in its current form.

7. PLOS authors have the option to publish the peer review history of their article (what does this mean?). If published, this will include your full peer review and any attached files.

Reviewer #1: No

Reviewer #3: No

---

## [Author Response · Author response to Decision Letter 1]

1 Apr 2024

Dear Reviewers':

Thank you for your letter and for the reviewers' comments concerning our manuscript entitled “The coupling coordination relationship between sports industry agglomeration and economic resilience in the Yangtze River Delta region." (Manuscript Number: PONE-D-23-37782). Those comments are all valuable and very helpful for revising and improving our paper, as well as providing important guidance for our further research. We have studied these comments carefully and have made corrections, which we hope will meet with approval. The words in blue are the changes I have made to the manuscript. The main corrections in the paper and the point-by-point responses to the reviewers' comments are as follows:

Response to Reviewer #1

1. However, I noticed a minor writing issue. For instance, there are several instances of "between the two" in this manuscript, which seems informal and incomplete. I suggest modifying it to "between the two systems" or using other pronouns or abbreviations.

Response: We thank the reviewer for this comment, and we strongly agree with the reviewers. At the same time, we have revised them to "between the two systems", words in blue are the change I have made in the file of “Revised Manuscript with Track Changes”.

2. Furthermore, as someone equally interested in system coupling and the Yangtze River Basin, I recommend that the author consider referencing and citing https://doi.org/10.3389/fevo.2023.1148868

Response: Thanks for reviewer’s comments, and we strongly agreed with that the entropy method does not require expert intervention and can effectively eliminate human subjective influence, so we cited the article of “Regional sustainability: Pressures and responses of tourism economy and ecological environment in the Yangtze River basin, China” in line 224-225. In addition, The order of the references 44 to 52 was also refined.

Special thanks to you for your good comments. We tried our best to improve the manuscript as your suggestion. And here we did not list all the changes but marked all the amends in revised paper.

We appreciate for your warm work earnestly, and hope that the correction will meet with approval. Thank you for the kind advice again. If you have any questions, please contact us without hesitate.

---

## [Editor Report · Decision Letter 2]

3 Apr 2024

The coupling coordination relationship between sports industry agglomeration and economic resilience in the Yangtze River Delta region

PONE-D-23-37782R2

Dear Dr.Hongyu Ma,

We’re pleased to inform you that your manuscript has been judged scientifically suitable for publication and will be formally accepted for publication once it meets all outstanding technical requirements.

Kind regards,

Lóránt Dénes Dávid, PhD

Academic Editor

PLOS ONE

Additional Editor Comments (optional):

The authors revised their paper, I accept this to publish.